# Impact of pain on functional status and quality of life in Jordanian women with breast cancer

**Nijmeh AL-Atiyyat**[1]*, **Hanan Fakhri Salem**[2], **Ammar Hisham Mustafa Hamam**[3]

**1** Department of Adult Health Nursing, Faculty of Nursing, The Hashemite University, Zarqa, Jordan,
**2** Department of Radiology-Oncology Al-Basheer Hospital, Amman, Jordan, **3** Department of Obstetrics and Gynecology, Al-Basheer Hospital, Amman, Jordan

* nijmeh@hu.edu.jo

**Data Availability Statement:** "The data sets generated during and analyzed during this study are not publicly available due to institutional copyright issues. Data are available from Hashemite University's Deanship of Scientific

## Abstract

### Background

Currently, breast cancer is the most prevalent type of cancer affecting women, and the number of newly diagnosed cases continues to increase both in Jordan and globally. Upon receiving a cancer diagnosis, the suffering experienced by patients intensifies as they grapple with the debilitating side effects that hinder their ability to carry out routine activities.

### Purpose

This study aims to assess the impact of cancer pain on functional status and quality of life (QOL) among Jordanian women with breast cancer.

### Methods

A descriptive cross-sectional design and structured interviews were conducted in the Oncology Unit at Al-Bashir Hospital with a sample of 150 eligible Jordanian women with breast cancer who provided data on pain and functional status, and QOL using four Arabic questionnaires (brief pain inventory, functional status SF-12 questionnaire, and quality of life index) to measure pain and functional status and QOL, descriptive statistics, Pearson correlation, and regression statistical test were utilized to analyze the study data.

### Results

A Significant positive correlation (r = 0.342) was found between cancer pain and functional status interference, a significant negative correlation (r = -0.134) between quality of life and functional status interference, and a significant negative correlation (r = -0.211) between pain and quality of life. Patients with higher intensity of cancer pain are more likely to experience low functional status and poor QOL.

Research by contacting Kalid A. F. AL-Zyoudi at khalid.zyoudi@hu.edu.jo. The authors confirm that other researchers would be able to access these data in the same manner as the authors and that the authors did not have any special access privileges that others would not have."

**Funding:** The author(s) received no specific funding for this work.

**Competing interests:** The authors have declared that no competing interests exist.

## Conclusion

Breast cancer patients experience alterations in their quality of life as a result of cancer-related pain, which has a detrimental impact on their ability to carry out daily activities and disrupts their functional abilities. Healthcare providers must take these alterations into account and effectively handle them. Understanding the relationship between cancer pain and breast cancer can aid nurses in managing symptoms and providing holistic care, ultimately improving the quality of life.

## Introduction

Breast cancer is the most prevalent form of cancer globally, with 7.8 million diagnosed in the past five years [1]. In Jordan, it accounted for 20.8% of total cases and 12.2% of deaths [2]. Cancer pain is a common and feared symptom among cancer patients, ranging from mild to severe intensity, with severity being more severe among those with significant burdens and undergoing treatments [3]. Pain from cancer can stem from the disease itself, therapies used, or complications. Breast cancer pain is common and can affect bones, joints, and lymph nodes. Tumor or cancer treatment, such as chemotherapy or surgery, can cause pain [4]. Cancer pain affects psychological, physical, and social aspects, making it crucial to understand common pain types when interacting with patients experiencing such pain [5].

Functional status is best defined as physical, psychological, social, spiritual, intellectual, and role-based activities an individual performs to meet his or her daily requirements in various aspects of life [6]. Functional status is a key factor in evaluating patients' quality of life within the first year of diagnosis. The use of intensive treatments like chemotherapy negatively impacts a patient's functional status, causing a decline in overall health regardless of the cancer type [7]. Additionally, fear and anxiety about the disease further exacerbate this decline [8]. Physicians generally have higher perceptions of health and well-being compared to patients, particularly among younger patients and those experiencing severe symptoms. Considering that functional status often correlates with age, it is expected that older patients receiving cancer treatment will experience a more pronounced decline in their functional abilities. In addition, their psychological and physical function is significantly impacted [9]. Gender, on the other side, women frequently report diminished performance in all types of activities [10].

Quality of life is the satisfaction and contentment people experience with their lives, considering factors like housing, employment, healthcare, education, and leisure activities. It is subjective and influenced by cultural and value systems [11]. Cancer and treatment modalities significantly impact the quality of life of cancer patients, impacting their ability to perform self-care, depression, and anxiety [12]. Proper management of cancer, including symptom management and treatment side effects, can significantly impact patients' quality of life. Pain, fatigue, and insomnia are the most intense symptoms, affecting quality of life negatively [13]. Advanced cancer patients are more likely to experience poorer quality of life due to aggressive treatment or side effects, making early detection crucial for enhancing quality of life [14]. Quality of life is crucial for cancer patients, with older patients prioritizing it over life expectancy, while younger patients often undergo aggressive therapy to increase survival time and quality of life [15]. Maximizing patients' quality of life enhances productivity, provides hope, encourages treatment adherence, improves community engagement, and provides hope when needed.

Various studies conducted in different countries have reported several contributing factors that can impact the quality of life of women diagnosed with breast cancer [16]. However, the factors that have been identified as influencing the quality of life in European and North American studies may not necessarily apply to Jordanian women with breast cancer. The discrepancy arises from variations in patients' objectives, anticipations, worries, and disparities in religious convictions, financial circumstances, healthcare, and social structures. Notably, the practices and perceptions of Jordanian women are primarily influenced by their religious beliefs [17]. Hence, comprehending the variables that affect the quality of life in Jordanian women diagnosed with breast cancer would have a beneficial effect on the standard of care given to these individuals and enhance the global understanding of the experiences of those affected by breast cancer.

Although various factors have been linked to the quality of life in women with breast cancer, the results have been inconclusive. Researchers have discovered that multiple factors such as age, marital status, medical complications, self-perceived health, sleep impairments, depression, and anxiety can have an impact on the quality of life for individuals with breast cancer [18, 19]. Nevertheless, other researchers have failed to discover substantial correlations among these variables [20, 21]. In addition to this controversial matter, we conducted this study among Jordanian women diagnosed with breast cancer due to a range of compelling factors: The high prevalence of breast cancer among Jordanian women and the lack of published research on the factors affecting their quality of life. Additionally, there is an incomplete consideration of potential factors, such as physical activity, that can impact their quality of life.

## Problem statement

Breast cancer is a widely spread cancer type in Jordan [2]. Cancer can worsen imbalances in roles and role failure, especially when symptoms and treatment complications are not managed effectively, leading to deterioration in patients' functional status and quality of life. Cancer pain significantly impacts patients' physical abilities and daily life, affecting their quality of life. However, the focus on cancer treatment and side effects often overlooks its impact on patients' functional status [22]. In Jordan, there is a lack of studies on the impact of cancer pain on patients' functional status. Addressing factors related to functional status declines could improve health outcomes and quality of life [23].

## Significance

Breast cancer is the leading cancer among women globally and in Jordan. Cancer pain is a common symptom and significantly impacts patients' functional status and quality of life.

To the researcher's knowledge, this is the first Jordanian study to explore the link between pain, functional status, and quality of life, filling a gap in existing studies. In our community, assessing an individual's health status is common by assessing their functional status and ability to perform their daily routine. Therefore, breast cancer, the disease itself, is well known across Jordan. It is time to shed light on further research targeting life improvement and functional status enhancement. Research should focus on life improvement and functional status enhancement to help these women continue their daily activities without limitations.

## Materials and methods

### Design

A quantitative descriptive, non-experimental correlational cross-sectional design was used to conduct this study.

## Setting

Jordan's health system offers oncology care across various public, military, private, and non-profit sectors. Al-Bashir Hospital, one of the country's oldest and largest governmental hospitals, serves diverse patients with over 1100 beds and over 12,000 outpatients daily.

## Sample and sampling

The study uses a non-probable convenience sample to target Jordanian women with breast cancer treated at Al-Bashir Governmental Hospital's oncology department.

## Sample size

In 2020, Jordan reported 2,403 breast cancer cases [24], with an estimated 3000 in 2023. Therefore, the descriptive study formula used to calculate the sample size was

$$N = \frac{z^2 \times \hat{p}(1-\hat{p})}{\varepsilon^2}$$

Confidence Level = 95%
The margin of Error ($\varepsilon$) = 0.05
Population Proportion ($\hat{p}$) = 5–10%
Population Size = 3000
Z score (z) = 1.96
According to the above formula, the required sample size was 134; however, we targeted 150 patients as the total sample size.

## Eligibility criteria

The study's participants were selected based on specific criteria to ensure their representativeness and external validity.

**Inclusion and exclusion criteria.** The study included adult Jordanian women aged 18 and above diagnosed with breast cancer, experiencing pain within the last two weeks, being cognitively intact, willing to participate, and read and write Arabic. Participants were excluded if they were women who underwent mastectomy or other recent surgery. Experiencing post-mastectomy pain is classified as acute, whereas this study investigates cancer pain, classified as chronic pain. Participants with cognitive impairment were also excluded.

## Procedure of data collection

A pilot study was conducted to modify the research protocol and ensure accurate completion of the questionnaire. A total of 10 patients were interviewed, with an average time of 20 minutes. Data was collected through structured interviews in outpatient and inpatient settings, with participants chosen based on inclusion criteria. An information sheet was distributed to participants, and consent was considered implicit when responding to the study's questions. The interview involved explaining medical terms and addressing body language to prevent bias. The interviews averaged 20–30 minutes, with data entered into an SPSS file concurrently. Data were collected From August 1st, 2022, to November 5th, 2022.

## Ethical consideration

The study was approved by Hashemite University's institutional review board and the Ministry of Health. The survey methodology ensured maintaining confidentiality; patients were informed of anonymity and confidentiality. No known personally identifiable information was

gathered. The responses were saved in the principal investigator's personal computers, secured with passwords. Subjects were guaranteed the option to take part in the survey voluntarily. Patients have consented verbally, and their responses to the research questions were considered implicit approval. Participants had the right to withdraw anytime without any consequences and contact the researcher for concerns. Data collection questionnaires were coded with numbers, securely stored, and used only for the study's purpose.

## Measurements

The short-form questionnaires measuring pain, functional status, and quality of life in Arabic were selected to minimize the potential subject burden. The datasheet provided demographic details such as age, marital status, education, cancer stage, number of children, and occupational status.

**Brief pain inventory-short form (BPI-SF).** The study used the Brief Pain Inventory-Short Form (BPI-SF) to measure pain levels and their impact on daily functioning. The questionnaire has two subscales: the pain severity scale and the pain interference scale. The severity scale includes questions about pain intensity, current treatments, and perceived effectiveness. The interference scale measures how pain interferes with activities, mood, walking, work, relationships, sleep, and life enjoyment. The Arabic version of the BPI-SF was chosen for reliability and validity [25], with Cronbach's alpha equal to 0.70 for this study. The Pain Severity and Interference scores represent the pain levels and impact on daily activities.

**Functional status SF-12 questionnaire.** The SF-12 is a shorter version of the SF-36 questionnaire. The SF-12 is a health-related quality-of-life questionnaire consisting of twelve questions that measure eight health domains to assess physical and mental health. Physical health-related domains include General Health (GH), Physical Functioning (PF), Role Physical (RP), and Body Pain (BP). The overall score of this questionnaire is given by two scores: a mental component scale (MCS) and a physical component scale (PCS). A score of 50 or less on the PCS-12 has been recommended as a baseline for identifying a physical condition, while a score of 42 or less on the MCS-12 may indicate 'clinical depression.' The Arabic version has been tested for reliability and validity in a study in Lebanon [26]. For this study, Cronbach's alpha for mental and physical components is 0.760, 0.703.

**Quality of life index–cancer version.** The quality-of-life index–cancer version measures the quality of life and consists of five subscales: health, functioning, social and economic, psychological and spiritual, and family. The questionnaire has 66 items, with scores ranging from 1 to 6. The Arabic version was tested for reliability and validity among Arabic-speaking patients [27], with a Cronbach's alpha of 0.703 for this study.

## Data analysis

The study's data was analyzed using the statistical package for the social science software SPSS (Version 22). The demographical data sheet was analyzed using descriptive statistics, mean, frequency, and standard deviation. BPI-SF questionnaire were analyzed by using descriptive statistics of the mean of items. While the Sf-12 questionnaire and QOL index cancer version questionnaire was analyzed by following the steps in the scoring manual of each.

Pearson's Correlation test was used to investigate the relationship between the three variables in the study: cancer pain, functional status, and quality of life. The Pearson correlation test is a statistical method for calculating the correlation coefficient between variables. The values vary from 1 to -1. A value of 0 indicates a lack of correlation between the variables.

A positive result indicates a direct relationship, while a negative result indicates an inverse relationship [28]. Correlation is a measure of the strength of the relationship between two

variables. It can be described using the guidelines proposed by Evans (1996) based on the absolute value of (r): 0.00 to 0.19 is considered "very weak," 0.20 to 0.39 is considered "weak," 0.40 to 0.59 is considered "moderate," 0.60 to 0.79 is considered "strong," and 0.80 to 1.00 is considered "very strong."

The statistical technique of "regression" was utilized to manage confounding variables, such as cancer stage. The method aims to assess whether the variables of interest statistically impact the dependent variable, the quality of life while considering other variables, specifically the cancer stage.

## Results

### Sample description

The study involved 150 women with breast cancer, with a mean age of 48.99 years, 74.0% being housewives, 26.7% primary school education, and 83.9% married. The majority had 3.58 children. The disease stage was third (60.0%), followed by stage four (34.7%), with stage two being the least common (5.3%) (Table 1).

### Cancer pain level

The study revealed an average pain level of 6.15 (SD = 1.60) among 150 patients, with the worst pain level being 8.19 (SD = 1.58) and the least being 1.61(SD = 1.64) within the last 24 hours, on the other hand, the average pain level during the interview was 1.88, (Table 2).

Notably, 88% of the respondents have a low level of pain severity, 6.7% have a moderate level of pain severity, and 5.3% have a high level of pain severity (Table 3).

Also, 84.7% of the respondents have a high pain Interference Score, 14.7% have a moderate pain interference score, and 0.7% have a low pain Interference Score (Table 4).

**Table 1. The demographic characteristics of the patients N = 150.**

| Variables | | Frequency (%) | *M (SD)* |
|---|---|---|---|
| Age (Years) | | | 48.99 (9.36) |
| Occupation | Housewife | 111 (74.0%) | |
| | Others | 33 (22.0%) | |
| | None | 6 (4.0%) | |
| Education | Illiterate | 21 (14.0%) | |
| | Primary | 40 (26.7%) | |
| | Middle | 29 (19.3%) | |
| | Highschool | 37 (24.7%) | |
| | Bachelor's degree | 21 (14.0%) | |
| | Postgraduate | 2 (1.3%) | |
| Marital Status | Single | 8 (5.3%) | |
| | Married | 134 (89.3%) | |
| | Divorced | 6 (4.0%) | |
| | Widowed | 2 (1.3%) | |
| Number of Children | | | 3.58 (2.07) |
| Cancer Stage | Second stage | 8 (5.3%) | |
| | Third stage | 90 (60.0%) | |
| | Fourth stage | 52 (34.7%) | |

**Table 2. Average pain level among Jordanian women with breast cancer N = 150.**

|  | N | Minimum | Maximum | Mean | Std. Deviation |
|---|---|---|---|---|---|
| Pain at its worst in the last 24 hours | 150 | 1 | 10 | 8.19 | 1.58 |
| Pain at its least in the last 24 hours | 150 | 0 | 8 | 1.61 | 1.64 |
| Average pain level | 150 | 0 | 9 | 6.15 | 1.60 |
| Current level of pain | 150 | 0 | 9 | 1.88 | 1.71 |

## Functional status

Results of functional status among patients with the mental component and physical component ranged widely; for the mental component subscale, the mean score was 34.7 (SD = 5.9) with arrange between 19.59 and 53.31, whereas the physical component subscale was 38.6 (SD = 5.1) with a range between 26.68 and 53.86. In order to establish a baseline for detecting a physical condition, a score of 50 or less on the PCS-12 has been suggested, whereas a score of 42 or less on the MCS-12 may indicate 'clinical depression' (Table 5).

75.3% of the respondents have a low level of functional status (physical), 19.3% have a moderate level of functional status (physical), and 5.3% have a high level of functional status (physical) (Table 6).

89.3% of the respondents have a low level of functional status (Mental), 8% have a moderate level of functional status (Mental), and 2.7% have a high level of functional status (Mental) (Table 7).

## Quality of life

Quality of life total score and subscales scores range from 0 to 30, with a minimum score of 11.71 and a maximum score of 30. The mean score of all subscales ranged between 17.80 (SD = 2.09) and 24.46 (SD = 2.63), whereas the total quality of life score had a mean of 20.57 (SD = 1.39), Noting the range of quality-of-life index scores range between (0–30) (Table 8).

35.3% of the respondents have a low quality of life, 32.7% have a moderate quality of life, and 32% have a high quality of life (Table 9).

## Interference of pain with the functional status

A frequency table was generated to identify different aspects of patients' lives that are highly affected by pain and cancer-related symptoms. From the results in Table 5 (Table 5), it was

**Table 3. Summary statistics for pain severity.**

| Pain Severity | N | % |
|---|---|---|
| Low | 132 | 88.0 |
| Moderate | 10 | 6.7 |
| High | 8 | 5.3 |
| Total | 150 | 100.0 |

**Table 4. Summary statistics for pain interference score.**

| Pain Interference Score | N | % |
|---|---|---|
| Low | 1 | 0.7 |
| Moderate | 22 | 14.7 |
| High | 127 | 84.7 |
| Total | 150 | 100.0 |

**Table 5. Average levels of functional status subscales, mental and physical components.**

|  | N | Minimum | Maximum | Mean | Std. Deviation |
|---|---|---|---|---|---|
| **Mental Component Scale (MCS)** | 150 | 19.59 | 53.31 | 34.75 | 5.92 |
| **Physical Component Scale (PCS)** | 150 | 26.68 | 53.86 | 38.68 | 5.13 |

evident that in breast cancer-related symptoms, the pain seemed to interfere with a different aspect of patients' lives; the scores ranged between 0–10; for instance, pain strongly interfered with their enjoyment of life (M = 8.61, SD = 1.6), sleep (M = 7.85, SD = 2.1), and mood (M = 7.71, SD = 2.1), general activity (M = 7.41, SD = 2.131) (Table 10).

## Pain, functional status, and quality of life

Correlation analysis was conducted to establish the association of pain, functional status, and quality of life among Jordanian women with breast cancer. With correlation coefficients (r) of 0.342, findings revealed a significant positive correlation between pain severity due to cancer and functional status interference of the body. This means pain severity is directly proportional to the interference of the functional status of the patient's body. The result further showed a statistically significant negative correlation between pain severity and the quality of life of cancer patients. This indicates that as the pain severity among cancer patients increases, the patient's quality of life decreases. Finally, the results also showed a statistically significant negative association between interference of the body's functional status due to cancer pain and the quality of life of cancer patients (r = -.134). These findings suggest that there is a correlation between pain, functional status, and quality of life among Jordanian women with breast cancer. Specifically, it indicates that as patients experience more interference in their functional status, their quality of life declines (Table 11).

In addition to the correlation analysis, a hierarchical linear regression analysis was conducted to measure the impact of pain severity (independent variable) on the quality of life of cancer patients (dependent variable) while considering the influence of cancer stage. Furthermore, this analysis was employed to manage other factors, particularly the cancer stage effectively.

The result indicates that the control variable, breast cancer stage, accounts for 2.2% of the variability in the quality of life among Jordanian women diagnosed with breast cancer. In

**Table 6. Summary statistics for functional status (physical) (N = 150).**

| Functional Status (Physical) | N | % |
|---|---|---|
| Low | 113 | 75.3 |
| Moderate | 29 | 19.3 |
| High | 8 | 5.3 |
| Total | 150 | 100.0 |

**Table 7. Summary statistics for functional status (mental) (N = 150).**

| Functional Status (Mental) | N | % |
|---|---|---|
| Low | 134 | 89.3 |
| Moderate | 12 | 8.0 |
| High | 4 | 2.7 |
| Total | 150 | 100.0 |

**Table 8. Average level of quality-of-life subscales and total quality of life score (N = 150).**

| | N | Range | Minimum | Maximum | Mean | Std. Deviation |
|---|---|---|---|---|---|---|
| Family Subscale **FAMSUB** | 150 | 12.50 | 17.50 | 30.00 | 24.46 | 2.63 |
| Psychological/Spiritual Subscale **PSPSUB** | 150 | 12.71 | 14.14 | 26.86 | 21.91 | 2.14 |
| Social and Economic Subscale **SOCSUB** | 150 | 13.14 | 14.14 | 27.29 | 21.70 | 2.22 |
| Health and Functioning Subscale **HFSUB** | 150 | 11.10 | 11.71 | 22.81 | 17.80 | 2.09 |
| Quality Of Life Index Total **QLI-T** | 150 | 8.55 | 16.17 | 24.72 | 20.57 | 1.39 |

addition, the severity of pain accounts for 3.4% of the differences in quality of life experienced by women diagnosed with breast cancer in Jordan. In addition, the P-value for the independent variable, pain severity (0.023), is below the 5% significance level. The data shows that there is a statistically significant relationship between the severity of pain and the quality of life among Jordanian women with breast cancer, as indicated by a beta value of -0.036 (Table 12).

## Effect of demographic data on pain, functional status and quality of life

The researcher used Pearson correlation coefficients and ANOVA to analyze study variables and demographic factors, examining differences based on age, number of children, education, marital status, and breast cancer. The quality-of-life index is negatively correlated with age and number of children, indicating that older patients have poorer quality of life (Table 13).

## Discussion

The study found a significant correlation between cancer pain, functional status, and quality of life. Higher pain levels were associated with lower functional status and quality of life. Functional status and quality of life are inversely correlated, with more functional interference affecting quality of life. Both factors measure overall health.

The study aligns with previous research by Anagnostopoulos et al., which found a strong correlation between pain and functional interference in cancer patients [29]. Sleep, enjoyment of life, and mood also significantly impacted moderate to severe pain. This aligns with a study by Schmidt et al. (2018), which suggests that sleep and decreased physical activity persist even in post-cancer diagnosis [30].

Cancer-related pain affects patients' sleep, leading to reduced activity and difficulty in daily life activities. This cycle is difficult to break without effective pain management. Functional impairments increase the likelihood of experiencing moderate to severe pain [31].

**Table 9. Summary statistics for quality of life (N = 150).**

| Quality Of Life Index | N | % |
|---|---|---|
| Low | 53 | 35.3 |
| Moderate | 49 | 32.7 |
| High | 48 | 32.0 |
| Total | 150 | 100.0 |

**Table 10. Interference of pain with functional status of patients in the last 24 hours (N = 150).**

| Variables | Range | Mean | Std. Deviation |
|---|---|---|---|
| Enjoyment of life | 0–10 | 8.61 | 1.601 |
| Sleep | 0–10 | 7.85 | 2.078 |
| Mood | 0–10 | 7.71 | 2.067 |
| General activity | 0–10 | 7.41 | 2.131 |
| Walking ability | 0–10 | 7.21 | 2.298 |
| Normal work | 0–10 | 6.88 | 2.326 |
| Relations | 0–10 | 6.83 | 2.254 |

Breast cancer patients have a low mean score in physical and mental components, indicating alterations in their health. Functional interference in life enjoyment, sleep, mood, and general activity is expected to result in low scores [26, 32].

Jordanian women, often working part-time or full-time jobs, face increased burdens of household chores, childcare, financial management, and errands, which can interfere with their functional functioning during cancer disease and treatment side effects.

## Cancer pain and quality of life

Cancer pain significantly impacts patients' quality of life, with a negative correlation. The more pain patients experience, the less quality of life they will experience. Studies have shown that increased pain decreases life quality (r = -0.671). Pain severity significantly impacts life quality in various domains, regardless of the cause or treatment of cancer [33, 34]. Pain is associated with lower life quality. Research on Jordanian breast cancer patients found that pain was strongly predicted and significantly associated with lower quality of life scores. This affects women's ability to perform daily tasks and meet societal roles, preventing them from participating in activities that bring happiness, relief, and joy, ultimately impacting their quality of life [35, 36].

## Pain, functional status, and quality of life alterations

The study found a significant correlation between pain, functional status, and quality of life, with patients experiencing more pain experiencing lower functional status and quality of life. However, few studies consistently found a consistent correlation between these variables.

A study among a smaller sample size of breast cancer patients exploring pain, fatigue, depression, and sleep disturbance investigated pain within subgroups at different times of treatment. Pain within all subgroups was associated with low functional status and quality of life [37]. Studies on functional status often focus on non-cancer patients with comorbidity, while older patients suffering from lower functional status, regardless of cancer, show a significant association between higher pain levels and lower functional status and poor life quality, regardless of various factors and settings [38].

**Table 11. Correlation coefficient of study variables (N = 150).**

| | Pain Severity | Functional Status Interference | Quality of Life |
|---|---|---|---|
| Pain Severity | 1 | | |
| Functional Status Interference | .342** | 1 | |
| Quality of Life | -.211** | -.134** | 1 |

**. Correlation is significant at 0.01 *[p value]* (2-tailed).

**Table 12. Result of regression for pain severity and quality of life controlling for cancer stage (N = 150).**

| Model | | Pain Severity. Quality of Life | | | Sig. |
|---|---|---|---|---|---|
| | | B | $R^2$ | $R^2$ Change | |
| | Breast Cancer stage | | .022 | | .073 |
| | Pain Severity | -.036 | .056 | .034 | .023 |

Dependent Variable: Quality of Life

Control Variable(s) (Breast Cancer Stage)

## Role of cancer stage, cancer pain in quality of life

Most women in this study sample were diagnosed with stage 3 and 4 breast cancer. The possible explanations are low education, as observed in this study. The majority finished only their primary and secondary education, as well as the fear of cancer diagnosis. Some women deny the need for medical care even when there is the presence of breast lump or shape alteration of the breast; even women who are aware of the self-breast examination and its importance do not commit to routine self-exam, on the other hand, the screening for breast cancer in Jordan are performed in two major centers, which result in a load of appointments that takes up to several months, financial issues also play a role as many women cannot afford mammograms in the private sector. As for this study, most women were non-working women with different lifestyles and priorities than women who have a job; during the interviews, some women stated that cancer-related pain and other side effects made them quit their jobs. Breast cancer patient's physical and mental balance can lead to functional imbalance, role failure, and quality of life decline. Proper assessment, management, and re-evaluation of interventions can prevent this [10].

## Strengths and limitations

The study explores the link between cancer pain, functional status, and quality of life among Jordanian breast cancer patients, a finding not previously reported. It suggests that understanding this association can improve healthcare providers' care and treatment, improving patient and caregiver quality of life encompassing a diverse patient population.

Several limitations need to be addressed. The cross-sectional study design, which provides data from a specific point in time, presents challenges in measuring incidence and causal

**Table 13. Pearson correlation coefficient between study variables and demographic variables (N = 150).**

| Variables | Patient's Age | Number of children |
|---|---|---|
| Social | -0.069 | -0.105 |
| Psychological | -0.068 | -0.136 |
| Family | 0.040 | 0.036 |
| Health and Functioning | 0.107 | 0.056 |
| Quality of Life | -.209* | -.206* |
| Functional Status (Physical) | -0.009 | 0.061 |
| Functional Status (Mental) | 0.046 | -0.045 |
| Pain Severity | 0.137 | -0.026 |
| Pain Interference Score | 0.011 | 0.113 |

**. Correlation is significant at 0.01 *[p value]* (2-tailed).

*. Correlation is significant at 0.05 *[p value]* (2-tailed).

inferences, necessitating longitudinal studies for better understanding. On the other hand, patients may refuse structured interviews due to pain, fatigue, or treatment procedures, known as respondent burden or survey fatigue. The study's results can be generalized to breast cancer patients in similar settings, though private hospital patients may provide different data under different circumstances. Other confounding variables, such as other comorbidities, should be considered for better correlation results. Using general scales and closed-ended questions, which may force respondents to select answers that do not adequately express their condition or viewpoint, is not ideal for determining the full impact on breast cancer patients.

## Recommendations and implications

The study found a significant link between cancer pain, functional status, and quality of life. Recommendations include raising public awareness, understanding the effects of breast cancer on women's life, and empowering them to manage their lives. Healthcare providers should strive to ensure a comprehensive assessment of cancer pain and functional status. This will enable them to provide the most appropriate care and improve their patient's quality of life. As nurses provide direct care to cancer patients and are the most familiar with their unique needs, they need to be aware of the cancer-related side effect and their association with deteriorating patient function and overall life quality to provide holistic care, which is the core of nursing.

Primary healthcare facilities must regularly provide health education programs that target the general population and those at risk of developing breast cancer.

Educators should reassess such topics in the nursing curriculum, teach the proper symptoms assessment, and include them in patient care.

For the researcher, Studies on other types of cancer can guide us to understand how cancer pain could produce similar or different effects on patients with other cancer types, which can help us better understand the association. In addition, such studies could further increase care effectiveness and give a deeper insight into different cancer treatments and prognoses.

Furthermore, a larger sample size can provide more accurate data and draw more reliable conclusions. The study's correlational nature suggests pain may not solely impact the quality of life, and further research is needed to explore other potential causes.

## Conclusion

Cancer pain alone is not enough to explain the overall poor life quality; the reduced functional status was an additional factor that had to be taken into consideration; by conducting this study, a suggestion of an association between cancer pain, functional status, and quality of life was found among Jordanian women with breast cancer, this study further highlights that cancer patients are significantly affected in terms of their day-to-day activities, further indicating the need for tailored interventions to address their pain and functional decline. Furthermore, these findings suggest that pain can drastically affect a patient's life quality, altering their ability to work, sleep, enjoy life, and maintain a positive mood. Therefore, it is essential to address pain to help individuals improve their quality of life by reducing the level of interference it has with their daily activities; this can involve taking a holistic approach and a combination of therapies, such as physical therapy, cognitive-behavioral therapy, pain management, and medications. Additionally, lifestyle modifications such as exercise, relaxation techniques, and adequate sleep can help to reduce pain and improve quality of life.

## Acknowledgments

The researchers express gratitude to all adult study participants.

## Author Contributions

**Conceptualization:** Nijmeh AL-Atiyyat.

**Data curation:** Hanan Fakhri Salem.

**Formal analysis:** Hanan Fakhri Salem, Ammar Hisham Mustafa Hamam.

**Investigation:** Nijmeh AL-Atiyyat, Ammar Hisham Mustafa Hamam.

**Methodology:** Nijmeh AL-Atiyyat.

**Project administration:** Nijmeh AL-Atiyyat.

**Supervision:** Nijmeh AL-Atiyyat.

**Validation:** Nijmeh AL-Atiyyat, Hanan Fakhri Salem, Ammar Hisham Mustafa Hamam.

**Writing – original draft:** Nijmeh AL-Atiyyat.

**Writing – review & editing:** Nijmeh AL-Atiyyat, Hanan Fakhri Salem, Ammar Hisham Mustafa Hamam.

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
