## [Decision Letter · Decision Letter 0]

10 Jun 2024

PONE-D-24-05458Impact of Pain on Functional Status and Quality of Life in Jordanian Women with Breast CancerPLOS ONE

Dear Dr. Al-Atiyyat,

Thank you for submitting your manuscript to PLOS ONE. After careful consideration, we feel that it has merit but does not fully meet PLOS ONE’s publication criteria as it currently stands. Therefore, we invite you to submit a revised version of the manuscript that addresses the points raised during the review process. Please submit your revised manuscript by Jul 25 2024 11:59PM. If you will need more time than this to complete your revisions, please reply to this message or contact the journal office at plosone@plos.org. Please include the following items when submitting your revised manuscript:A rebuttal letter that responds to each point raised by the academic editor and reviewer(s). You should upload this letter as a separate file labeled 'Response to Reviewers'.A marked-up copy of your manuscript that highlights changes made to the original version. You should upload this as a separate file labeled 'Revised Manuscript with Track Changes'.An unmarked version of your revised paper without tracked changes. You should upload this as a separate file labeled 'Manuscript'.If applicable, we recommend that you deposit your laboratory protocols in protocols.io to enhance the reproducibility of your results. Protocols.io assigns your protocol its own identifier (DOI) so that it can be cited independently in the future. For instructions see: https://journals.plos.org/plosone/s/submission-guidelines#loc-laboratory-protocols. Additionally, PLOS ONE offers an option for publishing peer-reviewed Lab Protocol articles, which describe protocols hosted on protocols.io. Read more information on sharing protocols at https://plos.org/protocols?utm_medium=editorial-email&utm_source=authorletters&utm_campaign=protocols.

We look forward to receiving your revised manuscript.

Kind regards,

Omar Mohammad Ali Khraisat, Associate Professor

Academic Editor

PLOS ONE

Journal Requirements:

4. Please upload a copy of Supporting Information Table. Supporting information which you refer to in your text on page 23-24.

Reviewers' comments:

Reviewer's Responses to Questions

**Comments to the Author**

1. Is the manuscript technically sound, and do the data support the conclusions?

Reviewer #1: Yes

Reviewer #2: Yes

2. Has the statistical analysis been performed appropriately and rigorously? 

Reviewer #1: Yes

Reviewer #2: Yes

3. Have the authors made all data underlying the findings in their manuscript fully available?

Reviewer #1: Yes

Reviewer #2: Yes

4. Is the manuscript presented in an intelligible fashion and written in standard English?

Reviewer #1: Yes

Reviewer #2: Yes

5. Review Comments to the Author

Reviewer #1: This is a very interesting study, yet, some minor changes need to be made. These include the following:

1. All abbreviations should be spelled out when they appear for the first time in the text or even in the tables.

2. Some editing is needed. For example, on page 5 line 119, instead of saying "to hit the light," it should say to shed the light. On line 120, an extra period should be deleted after the word enhancement. also page 7, the spelling of the months is not correct (line 156). On page 8, line 182, 4 concepts, not eight, or mention the eight concepts if they are 8. On page 11, line 228, it should read "with a range," not "arrange."

3. Additionally, lines 282-285, the paragraph on page 14, and the text lines 276-279, 276-291, require editing.

4. A mismatch was seen in the results section between the narrative and tables 3, 4, and 9. The results described in the text are different from those in these tables.

5. P values are better reported with the statistics where applicable. For example, the correlation matrix in Table 11, p-values obtained can be more convincing if they were included in the table with the r statistic.

Reviewer #2: Dear Editor

I hope this message finds you well. I am writing to provide my review of the manuscript entitled "Impact of Pain on Functional Status and Quality of Life in Jordanian Women with Breast Cancer." This study presents an important and timely exploration of how pain affects the functional status and quality of life in a specific population of breast cancer patients. Below, I have outlined my detailed feedback on the manuscript, incorporating some specific notes for improvement.

Summary of the Study

The manuscript addresses a significant topic by investigating the correlation between pain and its impact on the functional status and quality of life in Jordanian women with breast cancer. The authors employ a cross-sectional study design and use validated tools to measure pain, functional status, and quality of life, providing a comprehensive understanding of the patients' experiences.

Strengths

1. Relevance and Importance: The study highlights an under-researched area, contributing valuable insights into the experiences of Jordanian women with breast cancer.

2. Cultural Context: By focusing on Jordanian women, the study fills a gap in the literature regarding the cultural and regional aspects of pain management and quality of life in breast cancer patients.

Areas for Improvement

1. Literature Review: The literature review section could be expanded to include more recent studies and a broader range of perspectives on the topic. Incorporating more international research would provide a better context for the findings.

2. Clarity of Sentences: Line 83 contains sentences that are unclear and require further clarification for better understanding.

3. Settings and Sampling: There is a discrepancy regarding the settings from which the data were collected. While the manuscript mentions that information was collected from public, military, private, and nonprofit sectors, the sample and sampling section only references Al-Bashir Governmental Hospital. This needs to be clarified. Additionally, please specify the number of patients selected from each setting in Table 1.

4. Sample Size Calculation: Include a citation for the descriptive study formula used to calculate the sample size.

5. Data Analysis: The data analysis section would benefit from a more detailed explanation of the statistical methods used. Clarifying how the authors handled potential confounding variables would add to the study's credibility. Additionally, consider analyzing whether there are differences between groups from different settings regarding the study variables.

6. Results:

o Table 1: Include the number of patients selected from the various settings (governmental hospital, military, and other settings).

o Table 4: There is a contradiction between Line 220 and the data presented in the table regarding the percentage of respondents with low pain interference scores. This needs to be corrected.

o Line 227: Correct the phrase "with arrange" to ensure clarity.

7. Discussion and Implications: The discussion could be enriched by providing a more in-depth analysis of the implications of the findings for clinical practice and policy.

8. Formatting and Clarity: There are minor issues with the manuscript's formatting and clarity. For instance, some tables and figures could be better formatted to enhance readability. Additionally, ensuring consistent use of terminology throughout the manuscript would improve its overall coherence.

Recommendations

Overall, the manuscript presents valuable findings that could significantly contribute to the field of oncology and pain management. However, I recommend the authors address the aforementioned areas to strengthen the manuscript further. Specifically:

• Expand the literature review.

• Clarify discrepancies in the settings and sampling information.

• Provide citations for sample size calculations.

• Offer a detailed account of the statistical analysis methods.

• Enrich the discussion with practical implications and suggestions for future research.

• Address minor formatting and clarity issues.

I believe with these revisions, the manuscript has the potential to make a meaningful impact in the field. Thank you for the opportunity to review this important work.

Best regards

6. PLOS authors have the option to publish the peer review history of their article (what does this mean?). If published, this will include your full peer review and any attached files.

Reviewer #1: **Yes: **Hasan Al-Omran RN, OCN, PhD

Reviewer #2: No

---

## [Decision Letter · Decision Letter 1]

4 Jul 2024

Impact of Pain on Functional Status and Quality of Life in Jordanian Women with Breast Cancer

PONE-D-24-05458R1

Dear Dr.,

We’re pleased to inform you that your manuscript has been judged scientifically suitable for publication and will be formally accepted for publication once it meets all outstanding technical requirements.

Kind regards,

Omar Mohammad Ali Khraisat, Associate Professor

Academic Editor

PLOS ONE

Additional Editor Comments (optional):

Reviewers' comments:

Reviewer's Responses to Questions

**Comments to the Author**

1. If the authors have adequately addressed your comments raised in a previous round of review and you feel that this manuscript is now acceptable for publication, you may indicate that here to bypass the “Comments to the Author” section, enter your conflict of interest statement in the “Confidential to Editor” section, and submit your "Accept" recommendation.

Reviewer #1: All comments have been addressed

Reviewer #2: All comments have been addressed

2. Is the manuscript technically sound, and do the data support the conclusions?

Reviewer #1: Yes

Reviewer #2: Yes

3. Has the statistical analysis been performed appropriately and rigorously? 

Reviewer #1: Yes

Reviewer #2: Yes

4. Have the authors made all data underlying the findings in their manuscript fully available?

Reviewer #1: Yes

Reviewer #2: Yes

5. Is the manuscript presented in an intelligible fashion and written in standard English?

Reviewer #1: Yes

Reviewer #2: Yes

6. Review Comments to the Author

Reviewer #1: The authors have addressed all my comments properly. Yet, it is advisable that the whole manuscript be proofread for any typos, redundancy, or grammatical inconsistencies that I might have missed during the two rounds of review.

Reviewer #2: hope this message finds you well.

I have reviewed the revised version of your research paper. I am pleased to inform you that all my previous comments and suggestions have been thoroughly addressed. The revisions have significantly improved the clarity and quality of the manuscript.

Thank you for your diligent work in revising the paper.

Best regards,

7. PLOS authors have the option to publish the peer review history of their article (what does this mean?). If published, this will include your full peer review and any attached files.

Reviewer #1: **Yes: **Hasan Al-Omran RN, OCN, PhD

Reviewer #2: No

---

## [Editor Report · Acceptance letter]

16 Aug 2024

PONE-D-24-05458R1 

PLOS ONE

Dear Dr. Al-Atiyyat, 

I'm pleased to inform you that your manuscript has been deemed suitable for publication in PLOS ONE. Congratulations! Your manuscript is now being handed over to our production team.

Kind regards, 

on behalf of

Dr. Omar Mohammad Ali Khraisat 

Academic Editor

PLOS ONE